# The Effect of Quadruple Therapy with Polaprezinc or Bismuth on Gut Microbiota after *Helicobacter pylori* Eradication: A Randomized Controlled Trial

**DOI:** 10.3390/jcm11237050

**Published:** 2022-11-29

**Authors:** Dingkun Wu, Xinyue Li, Tingyuan Li, Wenbo Xie, Yujing Liu, Qinwen Tan, Wei Wu, Zhen Sun, Tingting Chen, Haidong Jiang, Jun Li, Junjie Qin, Yuqian Zhao, Wen Chen

**Affiliations:** 1Center for Cancer Prevention Research, Sichuan Cancer Hospital & Institute, Sichuan Cancer Center, School of Medicine, University of Electronic Science and Technology of China, Chengdu 610041, China; 2Department of Cancer Epidemiology, National Cancer Center/National Clinical Research Center for Cancer/Cancer Hospital, Chinese Academy of Medical Sciences and Peking Union Medical College, Beijing 100021, China; 3Institute of Life Science and Green Development, College of Life Sciences, Hebei University, Baoding 071000, China; 4School of Life Sciences, Jilin University, Jinlin 130021, China; 5National Children’s Medical Center, Children’s Hospital of Fudan University, Shanghai 201101, China; 6College of Health Management, China Medical University, Shenyang 110122, China; 7Department of Gastroenterology, Jilin People’s Hospital, Jilin 130021, China; 8West China School of Public Health and West China Fourth Hospital, Sichuan University, Chengdu 610065, China; 9Cancer Prevention and Treatment Office, Yanting Cancer Hospital, Mianyang 621699, China; 10Shenzhen Promegene Technology Co., Ltd., Shenzhen 518038, China

**Keywords:** *Helicobacter pylori*, polaprezinc, bismuth, gut microbiota, 16S rRNA gene sequencing

## Abstract

Background: Quadruple therapy with polaprezinc provided an alternative to *Helicobacter pylori* eradication; however, the effect on gut microbiota remains uncertain. This study aims to identify whether polaprezinc-containing quadruple therapy causes adverse microbiota effects among asymptomatic adults, compared with bismuth therapy. Methods: This was a randomized control trial. One hundred asymptomatic *H. pylori*-infected adults were randomly (1:1) assigned to two treatment groups (polaprezinc-containing therapy, PQT; or bismuth-containing therapy, BQT). Fecal samples were collected from subjects before and 4–8 weeks after therapy. Samples were sequenced for the V4 regions of the 16S rRNA gene. Results: The relative abundance of the three dominant bacterial phyla (Bacteroidota, Firmicutes, and Proteobacteria) accounted for more than 95% of each treatment group. The alpha diversity between eradications that succeeded and those that failed had no significant difference (*p* > 0.05). After successful eradication, the alpha diversity in the BQT group decreased in comparison with the baseline (*p* < 0.05). Subjects who were successfully eradicated by BQT showed considerably lower alpha diversity indices than those of the PQT at follow-up (*p* < 0.05). The abundance of *Parasutterella* in subjects who were successfully eradicated by PQT was four times greater than that of BQT (*q* < 0.05). Conclusion: A 14-day PQT may be superior to BQT in maintaining short-term gut microbiota homeostasis after *H. pylori* treatment. Our findings preliminarily provide evidence of the short-term impacts of the gut microbiota after PQT treatment of *H. pylori* infection.

## 1. Introduction

*Helicobacter pylori* (*H. pylori*) infects nearly half of the world’s population and causes gastrointestinal diseases such as chronic gastritis, peptic ulcers, gastric cancer (GC), and mucosa-associated lymphoid tissue (MALT) lymphoma [1,2,3]. Over 768,000 deaths due to GC occurred in 2020, ranking as the 4th leading cause of cancer death [4]. The attributable risk of *H. pylori* on gastric cancer was estimated to be 78%, which was considered the most significant risk factor for the development of gastric cancer [5]. It was reported that 1–10% of the infected individuals would develop gastric or duodenal ulcers, and 0.1–3% of GC patients were caused by *H. pylori* infection [6]. Thus, the International Agency for Research on Cancer (IARC) identified *H. pylori* as a class I carcinogen in 1994 [7]. *H. pylori* eradication programs for gastric cancer prevention have been shown to benefit high-risk populations [8]. As a result, clinical guidelines recommend adopting the *H. pylori* “screen and treat” strategy in high-risk populations for GC worldwide [9,10,11].

The Maastricht 2-2000 Consensus Meeting recommended the use of standard triple therapy as the first-line treatment strategy for *H. pylori* eradication in 2002 [12]. It combines a proton pump inhibitor (PPI) plus clarithromycin, together with amoxicillin or metronidazole. In just over a decade, however, increasing antibiotic resistance has dramatically reduced the eradication rate of triple therapy for *H. pylori*. Due to the excellent efficacy of *H. pylori* eradication, bismuth-containing quadruple therapy (BQT), which comprises PPI, bismuth, and two antibiotics, is currently recommended as first-line therapy [9,13]. However, mounting evidence suggested that eradication therapy might cause dysbiosis of the gut microbiome including the reduction in microbiota community diversity and bacterial taxonomic changes [14,15,16,17,18]. One study found that the patients eradicated by BQT had gut microbiota dysbiosis for more than 1 year [19]. Additionally, the dysbiosis of the gut microbiome significantly affected human pathophysiology and was correlated with other diseases [20]. Therefore, it is crucial to explore the effect of *H. pylori* eradication therapy on the microbiota and the promising therapeutic strategies to maintain gut microbiota homeostasis. Currently, scientists are exploring other promising therapies for *H. pylori* infection as well [21,22,23,24]. One of them is polaprezinc-containing quadruple therapy (PQT). Polaprezinc has antioxidative effects, scavenging urease and monochloramine and inhibiting *H. pylori*-associated inflammation [25,26,27,28]. It can be used as a gastric mucosal protectant to promote the healing of peptic ulcers and eradicate *H. pylori*. Previous studies confirmed the therapeutic efficacy of PQT in Asian countries, without causing serious adverse events [24,29]. Hence, we conducted a study based on a randomized clinical trial to research the prospective impact of quadruple therapies with polaprezinc or bismuth on gut microbiota.

## 2. Methods

### 2.1. Study Design and Population

This study was a randomized clinical trial following the CONSORT guidelines. The study design was previously published [30]. One hundred asymptomatic adults aged 40 to 69 years were enrolled at Yanting Cancer Hospital from December 2019 to August 2020. Diagnosis of *H. pylori* infection was by ^14^C-urea breath test(^14^C-UBT). Subjects were enrolled according to the inclusion and exclusion criteria (Appendix A) and were given a 14-day quadruple anti-*H. pylori* treatment, either the PQT group (esomeprazole 20 mg, amoxicillin 1 g, clarithromycin 500 mg, polaprezinc 75 mg, twice daily) or the BQT group (esomeprazole 20 mg, amoxicillin 1 g, clarithromycin 500 mg, bismuth potassium citrate 220 mg, twice daily). The subjects were randomly allocated (1:1) to the two groups using a computer-generated random sequence. Allocation was concealed in opaque envelopes until the beginning of the intervention. Self-reported gastrointestinal symptoms were evaluated on day 7 and day 14 of the medication. All subjects received a second ^14^C-UBT (to confirm the eradication efficacy), a physical examination, an endoscopic examination, and fecal sample collection four to eight weeks after treatment. All subjects were offered informed consent. All subjects were not allowed to take any drugs that affect the flora during the trial. This study was approved by the Ethics Committee of Sichuan Cancer Hospital and was registered at the Chinese Clinical Trial Registry (ChiCTR1900025800, 9 September 2019). All methods were carried out in accordance with relevant guidelines and regulations.

Trained medical staff collected the demographic data through a face-to-face interview. Two experienced gastroenterologists performed endoscopic examinations using video endoscopes (Olympus) according to the guidelines for cancer screening and early diagnosis and treatment in China.

### 2.2. Fecal Sample Collection

On the day of the enrolment (before treatment) and during the follow-up (4–8 weeks after treatment), fresh fecal samples were collected from all subjects. The subjects obtained their fecal samples at the hospital, following the standard procedures, and returned them to the medical staff. All fecal samples were transferred and stored at −80 °C in an ultra-low temperature refrigerator until further processing.

### 2.3. DNA Extraction, Amplification, and Sequencing

The genomic DNA from fecal samples was extracted by MOBIO PowerSoil^®^ DNA isolation Kit 12888-100 (QIAGEN, Shanghai, China) and stored in Tris-EDTA buffer (Sigma-Aldrich, Shanghai, China) at −80 °C until use. The V4 region of the microbial 16S rRNA gene was amplified using the universal primers: forward primer 515F (5′-GTGCCAGCMGCCGCGGTAA-3′) and reverse primer 806R (5′-GGACTACHVGGGTTCTAAT-3′), which were fused with the barcode sequence to differentiate samples. The total PCR amplification reaction system is 50 μL, which includes 1 μL forward primer and 1 μL reverse primer (10 μM), 1 μL of template DNA, 4 μL of dNTPs (2.5 mM), 5 μL of 10 × EasyPfu Buffer (Abbexa, Shanghai, China), 1 μL of Easy Pfu DNA Polymerase (Abbexa, Shanghai, China) (2.5 U/μL), and 1 μL of double-distilled water. The following PCR thermal cycling conditions were used: an initial denaturation step at 95 °C for 5 min, followed by 30 cycles of denaturation at 94 °C for 30 s, annealing at 60 °C for 30 s, and extension at 72 °C for 40 s, with a final extension step at 72 °C for 4 min. Amplicons in the size range of 300–350 bp from each sample were obtained. Following the manufacturer’s instructions, the authors quantified amplicons with the Quant-iT PicoGreen dsDNA Assay Kit (ThermoFisher/Invitrogen cat. no. P11496, Shanghai, China). The amplicon library was combined in equal amounts and subsequently quantified (KAPA Library Quantification Kit KK4824, KAPA Biosystems, Beijing, China). High-throughput sequencing on the Illumina MiniSeq platform at Promegene Co., Ltd. (Shenzhen, China) was performed to generate 150 bp paired-end reads (excluding the primer sequences) for each sample.

### 2.4. Sequencing Data and Microbiota Diversity Analysis

The raw data were processed using QIIME2 [31] (https://qiime2.org/, accessed on 21 September 2021). A truncation of all reads at the 150th base with a median Q score > 20 was performed to avoid sequencing errors at the end of the read. The DADA2 algorithm was used to clear noisy sequences and impurities from sequence data [32]. The denoised pairwise end reads were concatenated and the maximum mismatch parameter was set to 2 bases. The representative sequences (i.e., the features) were defined as 100% similar merged sequences. For convenience, the term “operational taxonomic unit (OTU)” was used as an alternative to “feature” throughout the article. Afterward, the taxonomy of the bacterium was identified from chimera-filtered compared with the SILVA [33] (version 128, http://benjjneb.github.io/dada2/training.html, accessed on 21 September 2021) 16S rRNA database. They were assigned to taxa (domain, kingdom, phylum, class, order, family, genus, and species) using a trained Naive Bayes classifier.

The alpha diversity of gut microbiota was estimated using Shannon’s diversity index, observed_otus, Faith’s phylogenetic diversity (a qualitative measure emphasizing the genetic basis and sequence richness), and Pielou’s evenness. The beta diversity was evaluated via Bray–Curtis distances between samples. The principal coordinate analysis (PCoAs) was used for visualization.

### 2.5. Statistical Analyses

All statistical analyses were performed using SPSS17.0 (SPSS, Chicago, IL, USA) and R software (version 3.6.1; https://www.r-project.org/, accessed on 12 October 2021) on PC. Intention-to-treat (ITT) and per-protocol (PP) analyses were used to evaluate the efficacy of *H. pylori* eradication. Patients who met the criteria were randomized and took at least one dose of medication after enrolling were included in the ITT population. The PP population included patients who completed the designated therapy for this protocol. Normally distributed data were shown as the mean ± standard deviation (SD), whereas non-normal data were shown as the median and interquartile range. The Student’s *t*-test was used to compare normal continuous data, while the Mann–Whitney U test was used for non-normal data distribution. The chi-square test was used to make statistics on categorical variables, which were shown as percentages. The level of significance was set as a 2-sided *p* value less than 0.05. *p*-values ≤ 0.05 were considered significant.

The paired Wilcoxon signed-rank tests were performed for the paired samples (within-group) comparisons of alpha diversity indices and taxa differences at baseline and follow-up. Between-group differences at baseline and end of treatment were assessed using the unpaired Wilcoxon signed-rank test. A PCoA was performed to describe the differences in beta diversity. *p*-values were calculated by the permutational multivariate analysis of variance (PERMANOVA) test, which was performed with the adonis function of the R vegan package and generated based on 1000 permutations. *p*-values adjusted for correction of multiple comparisons by Benjamini and Hochberg’s false discovery rate (FDR) approach and corresponding *q* values < 0.05 were considered significant and calculated based on the R fdrtool package.

## 3. Results

### 3.1. Overview of the Study Population

One hundred subjects with *H. pylori* infection were involved in the study, with 49 being assigned to the PQT group and 51 to the BQT group. Among them, 81 subjects were followed-up, and 61 subjects completed sample collection at follow-up (Figure 1). There were no significant differences between the two groups in age, gender, body mass index (BMI), smoking habits, alcohol consumption, and gastroscopy results (all *p* > 0.05) (Table 1). There were no significant differences in age, gender, BMI, education level, or gastroscopy results between the completed (61 subjects) and incomplete (39 subjects) fecal sample collection populations (all *p* > 0.05); nevertheless, a significant difference in smoking habits was observed (*p* < 0.05) (Appendix A). A total of 5.45 million clean reads were obtained from the sequencing data of fecal samples, with an average of 44,673 reads per sample. The 61 subjects who completed sample collection were divided into four subgroups: the PQT successful eradication group (15 subjects), the BQT successful eradication group (21 subjects), the PQT failed treatment group (5 subjects), and the BQT failed treatment group (5 subjects). Fifteen subjects were delayed for sample collection due to the COVID-19 pandemic and were analyzed separately.

### 3.2. The Difference in Eradication between the PQT and BQT Treatment Groups

In the ITT analysis, the eradication levels of the PQT and BQT groups were 61.22% and 70.59%, respectively. According to PP analysis, the eradication rates of the PQT and BQT groups were 75.0% and 87.80%, respectively. There were no statistical differences in eradication rates between groups (Table 2, *p* > 0.05). No differences in *H. pylori* eradication between the 20 subjects who completed the study but did not return fecal samples and the 61 subjects who completed the study were observed (80.00% vs. 81.97%, *p* > 0.05). One subject (2%) in the PQT group developed skin allergies during the medication and dropped out. In the BQT group, one subject (2%) had an increase in total bilirubin (from 30.9 μmol/L at baseline to 43.8 μmol/L at follow-up), twice the normal level after treatment. Self-reported gastrointestinal symptoms during treatment included nausea, bloating symptoms, and so on (Appendix A), but no significant statistical differences were observed between the two groups (*p* > 0.05).

### 3.3. Alpha Diversity Alteration on Gut Microbiota

The median follow-up time interval was 35 (29–53) days for the PQT group and 43 (34–53) days for the BQT group, with no significant difference (*p* = 0.22). Shannon’s diversity, Pielou’s evenness, observed_otus, and Faith’s phylogenetic diversity indices of the paired samples for the four subgroups (PQT successful eradication group, BQT successful eradication group, PQT failed treatment group, and the BQT failed treatment group) are shown in Figure 2 and Appendix A. Compared with the baseline, the alpha diversity significantly decreased in the subjects who were successfully eradicated by BQT (*p* < 0.05). Compared with the baseline, there was no statistical difference in the subjects who failed to be eradicated by either the PQT or the BQT treatment, as well as those who were successfully eradicated by PQT (Figure 2A,B and Appendix A) (*p* > 0.05). At follow-up, the alpha diversity indices among successfully eradicated subjects by BQT were significantly lower than those of PQT (Figure 2C,D and Appendix A) (*p* < 0.05).

### 3.4. Beta Diversity Alteration on Gut Microbiota

The beta diversity at follow-up showed no statistical difference between the paired samples of the PQT and BQT groups compared with baseline, irrespective of the eradication outcome (Figure 3A,B and Appendix A). It indicated that the microbiome structure at follow-up showed no significant variety or was restored after treatment. The beta diversity among the four subgroups is shown in Figure 3. At follow-up, a significant difference between the PQT successful eradication group and the BQT successful eradication group was observed (*p* < 0.05). However, there was no significant difference in the beta diversity after failed treatment between the PQT and BQT groups (Figure 3C,D).

### 3.5. Differences in Gut Microbiota Taxa between the PQT and BQT Treatments

To further evaluate the specific changes in gut microbiota, we analyzed the taxonomic composition and relative abundance of bacterial taxa identified by 16S rRNA sequencing in the four subgroups. The relative abundance of the three dominant bacterial phyla (Bacteroidota, Firmicutes, and Proteobacteria) accounted for more than 95% of each group at the phylum level. At the genus level, the three dominant bacterial genera were *Prevotella*, *Bacteroides*, and *Faecalibacterium*, accounting for more than 51% of each group (Figure 4A,B). Although there were only five subjects in the failed treatment group of BQT, abnormal enrichment of the *Prevotella* genus was observed at baseline and follow-up. The dominant bacterial genera, *Prevotella*, *Bacteroides*, and *Faecalibacterium*, were similar to the baseline in the PQT and BQT groups after treatment.

There were 12 genera of taxa differences between the successful eradication of PQT and BQT after comparing the changes in specific taxa at follow-up (*p* < 0.05). However, correcting by FDR, we only observed that the relative abundance of *Parasutterella* (*q* < 0.05) in subjects successfully eradicated by PQT was four times greater than in subjects successfully eradicated by BQT (Table 3).

### 3.6. Diversity Alteration in the Gut Microbiota of the 15 Subjects Who Delayed the Fecal Sample Collection

The median follow-up time interval of the PQT and BQT groups was 74 (72–109) days and 63 (62–90) days, respectively, without significant differences (*p* = 0.19). Six subjects in the PQT group were successfully eradicated, six subjects in the BQT group were successfully eradicated, and three subjects in the PQT group failed the eradication. The alpha diversity at follow-up revealed that these three groups had no statistical difference compared with the baseline (Appendix A). At the same time, the between-group variance in alpha diversity of these three groups at baseline and follow-up was not observed (Appendix A). In addition, no difference in beta diversity was found in the same analysis process (Appendix A).

## 4. Discussion

In this study, we conducted a prospective, randomized trial to examine the effect of quadruple therapy with polaprezinc or bismuth on the gut microbiota after *H. pylori* eradication. The results demonstrated that the alpha diversity decreased compared with baseline for subjects who were successfully eradicated by BQT and was lower at follow-up than that of PQT. The abundance of the *Parasutterella* genus notably changed in the gut flora between the PQT and the BQT treatments. The findings implied that PQT might provide potential benefits in maintaining gut microbiota homeostasis.

A persistent *H. pylori* infection caused chronic inflammation of the gastric mucosa, as well as changes in cellular signaling associated with gastric cancer [34,35,36], cellular metabolism regulation [37], and autophagy, which allowed the bacteria to escape eradication therapy and led to infection recrudescence after treatment [38,39,40]. As a result, antibiotic resistance may increase. It was reported that bismuth inhibits metallo-lactamase activity [41], disrupts the central carbon metabolism of *H. pylori* [42], and enhances the effectiveness of growth-dependent antibiotics [43]. A study found that eradication of *H. pylori* with bismuth-containing quadruple therapy causes short-term dysbacteria [44]. Moreover, the bismuth agent is a kind of heavy metal, which could cause adverse effects on human health after long-term application. Polaprezinc, a chelate compound composed of zinc and L-carnosine, may replace the nickel ion of the active center of urease to inactivate the urease, causing the inhibition of the growth of *H. pylori* [45]. Furthermore, zinc can directly modulate the host’s immune response to infection by regulating the innate immune response in the host [46]. A previous single-center, small sample size study conducted in Japan determined that polaprezinc combined with triple therapy increased the *H. pylori* eradication rate from 77.4% (24/31) to 94.3% (33/35) [29]. Subsequently, a randomized, controlled, multicenter study confirmed that polaprezinc combined with omeprazole, amoxicillin, and clarithromycin can improve the *H. pylori*-infection cure rate from 62/101 (61.4%, triple therapy alone) to 86/106 (81.1%) in the Chinese population [24]. It has been demonstrated that the eradication rate of *H. pylori* in clarithromycin low-resistance areas of Europe was up to 90% [47]. The high prevalence of clarithromycin resistance limited the eradication effect [13]. A systematic review showed that clarithromycin resistance was up to 35% in southwestern China [48]. In our study, the eradication rate of polaprezinc-containing quadruple therapy was non-inferior to conventional bismuth-containing quadruple therapy (75.0% vs. 87.8%). Taken together, for *H. pylori* infection patients with contraindications to bismuth, PQT can be selected as an alternative eradication regimen.

In this study, the performance of PQT in maintaining the gut microbiota homeostasis was observed and showed no adverse increase effect after treatment. Previous multiple timepoint follow-up studies [19,44] reported that the microbial diversity would decrease at the end of treatment or in the second week, begin to recover at 8 weeks, and then return to baseline levels for 6 months or longer. A single follow-up study conducted by Olekhnovich [14] suggested that the alpha diversity significantly decreased on day 0 or within 2 days and had the tendency to reshape after *H. pylori* eradication therapy. Liou et al. reported [19] that the alpha diversity of the BQT therapy was significantly reduced compared with that of the triple therapy at 8 weeks after treatment. Similar to the previous findings, the alpha diversity of the BQT therapy in this study decreased compared with the baseline nearly 6 weeks after eradication. We found that the alpha diversity of the PQT therapy was significantly increased compared to that of the BQT therapy at follow-up. 

Similar to previous reports, the host symbiotic microbiome contained Bacteroidetes, Firmicutes, and Proteobacteria in the gut microbiota of all subjects [15,44]. Specific microbiota alteration in the gut microbiome varies with *H. pylori* eradication therapy. Increased *Bifidobacterium* was found in the gut microbiota following successful eradication with 10-day BQT therapy [49]. Another study showed a significant reduction in the relative abundance of *Bifidobacterium adolescentis*, while *Enterococcus faecium* levels increased 0 or 2 days after the 14-day BQT treatment [14]. The abundance of some genera in the Firmicutes and Proteobacteria was lower than those at baseline after one year of 14-day reverse hybrid therapy treatment, including *Brochothrix*, *Lysinibacillus*, *Solibacillus*, *Enhydrobacter*, *Psychrobacter*, and *Pseudomonas* [50]. After 2 weeks of treatment with *S. boulardii*, the abundance of Bacteroides (*Prevotella*) and Clostridia (*Lachnospira* and *Ruminococcus*) reduced, and the abundance of Gammaproteobacteria (*Escherichia* spp. and another *Enterobacteriaceae* OTU) increased compared with the conventional triple therapy [51]. In a multicenter randomized controlled trial, the researchers found *Shigella*, *Klebsiella*, and *Streptococcus* were enriched after BQT therapy [52]. Our results indicated that the relative abundance of *Parasutterella* in subjects successfully eradicated by PQT was four times greater than that of BQT. *Parasutterella* isolates have been considered a core microbe of the human and mouse gut microbiota [52]. Significant reductions in *Parasutterella* abundance can be observed in mice treated with a high-fat diet (HFD) [53]. Similarly, *Parasutterella* of gut microbiota was negatively correlated with indicators of body weight and serum lipids in obese patients [54]. However, this finding was not consistent in different diseases. For instance, the abundance of *Parasutterella* in fecal specimens from the irritable bowel syndrome (IBS) mice model was significantly higher than that in normal controls [55]. Although *Parasutterella* showed inconsistent findings in the animal model studies above, a negative association between *Parasutterella* and obesity was observed in human studies. It is suggested that *Parasutterella* may be beneficial to human health in some specific host environments and further studies are still needed to confirm the association between *Parasutterella* and intestinal health.

The differential taxa microbiome of the paired samples had no significant change in our study. According to a previous study, triple therapy produces fewer disorders of the gut microbiota, while BQT slowed the restoration of gut microbiota diversity after *H. pylori* eradication [19]. There is currently limited evidence that zinc is associated with gut microbiota in animal studies [56,57,58]. The Chao1 index and observed species richness showed that the Zn-deficient broiler chicken (*Gallus gallus*) model had significantly lower α diversity and species richness in cecal microbiota than the control group [58]. A recent study has shown that formula feeding for children with Zn deficiency leads to a decrease in the abundance of several genera, such as *Escherichia*, *Streptococcus,* and *Bacteroides* [59]. Another study found that a short-term (4-week) low-Zn diet (0 mg/kg) and a long-term (8-week) high-zinc diet (150 mg/kg) had obvious negative effects in a mouse model, primarily through changing the gut bacterial composition (e.g., *Lactobacillus reuteri* and *Akkermansia muciniphila*) [60]. Interestingly, oral administration of zinc oxide nanoparticles did not change alpha diversity but interfered with gut microbial composition [61]. A higher abundance of *Lactobacillus* was observed in the ileal digesta, while the abundance of *Streptococcus*, *Escherichia shigella*, *Actinobacillus*, and *Clostridium sensu stricto 6* was significantly decreased in chitosan-chelated zinc group piglets. These results illustrate that CS-Zn treatment could help regulate the composition of ileal flora and improve ileal health [62]. Overall, evidence of the efficacy of physiological zinc supplementation in promoting a healthy supportive effect of normal gut microbiota seems to be growing. Based on the findings of the prior report and the current study, we propose that PQT may cause less disruption to the gut microbiota.

To the best of our knowledge, this is the first randomized controlled trial comparing the effects of polaprezinc and bismuth therapy for *H. pylori* eradication on gut microbiota. Although this is a small sample size, single-center, single-visit study without negative fecal control, good quality was maintained through the whole process of study, e.g., high compliance, sample collection, and laboratory operation. 

In conclusion, a 14-day PQT may be superior to the BQT in maintaining short-term gut microbiota homeostasis after *H. pylori* treatment. Our findings preliminarily provide evidence of the short-term effects of the gut microbiota after PQT treatment of *H. pylori* infection. Further validation studies with multiple visits, multiple centers, and large sample sizes are required.

## Figures and Tables

**Figure 1 jcm-11-07050-f001:**
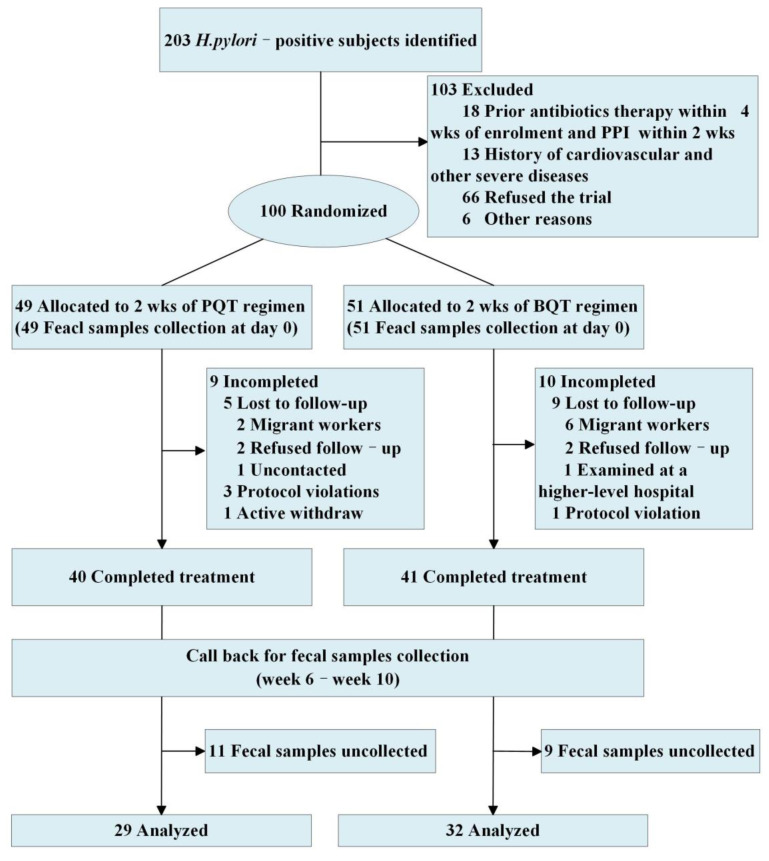
CONSORT flow diagram of this study.

**Figure 2 jcm-11-07050-f002:**
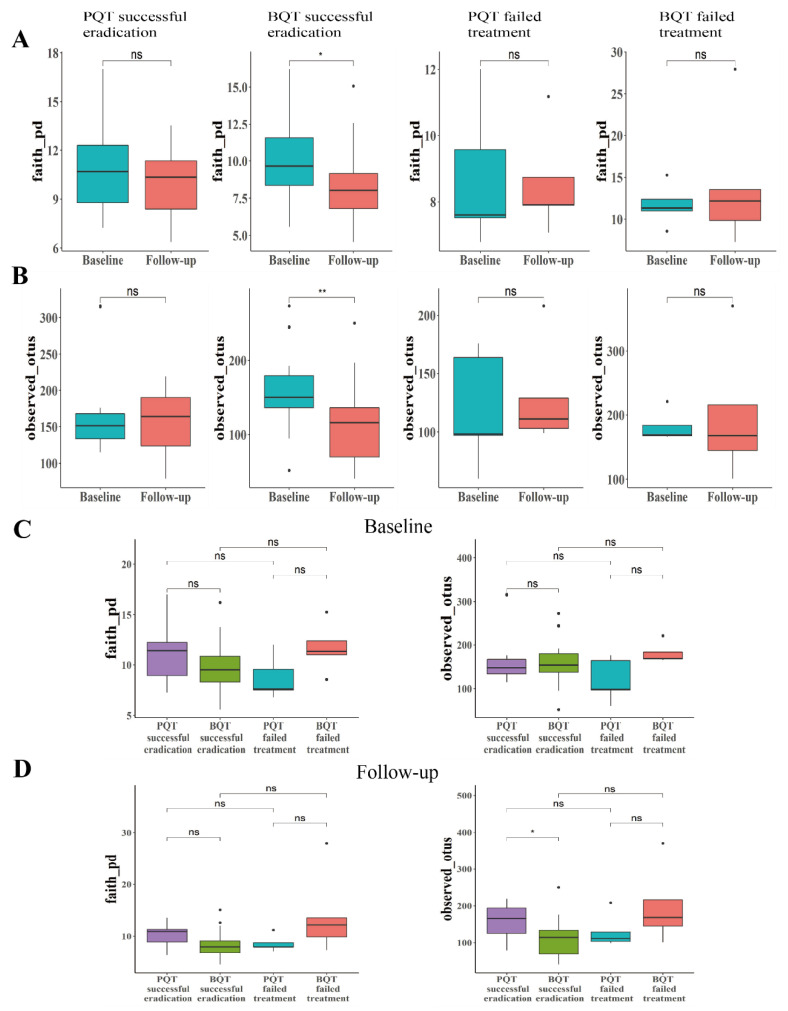
The comparisons of alpha diversities among the four groups. Faith pd (**A**) and observed otus (**B**) indices represent the alpha diversity changes from baseline to the end of treatment. The comparison of intergroup alpha diversities differences at baseline (**C**) and follow-up (**D**). *, *p*-values < 0.05, ** *p*-values < 0.01.

**Figure 3 jcm-11-07050-f003:**
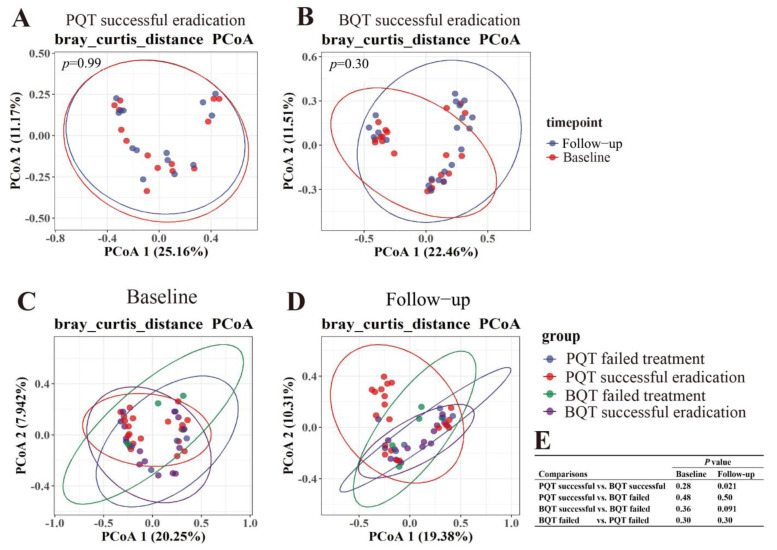
Comparisons of beta diversities. Beta diversity (PCoA) was compared with the baseline in PQT successful eradication group (**A**) and BQT successful eradication group (**B**). Beta diversity was compared among the four groups at baseline (**C**) and follow-up (**D**). (**E**): *p*-adjusted values of the multiple comparisons. PQT, polaprezinc-containing therapy; BQT, bismuth-containing therapy.

**Figure 4 jcm-11-07050-f004:**
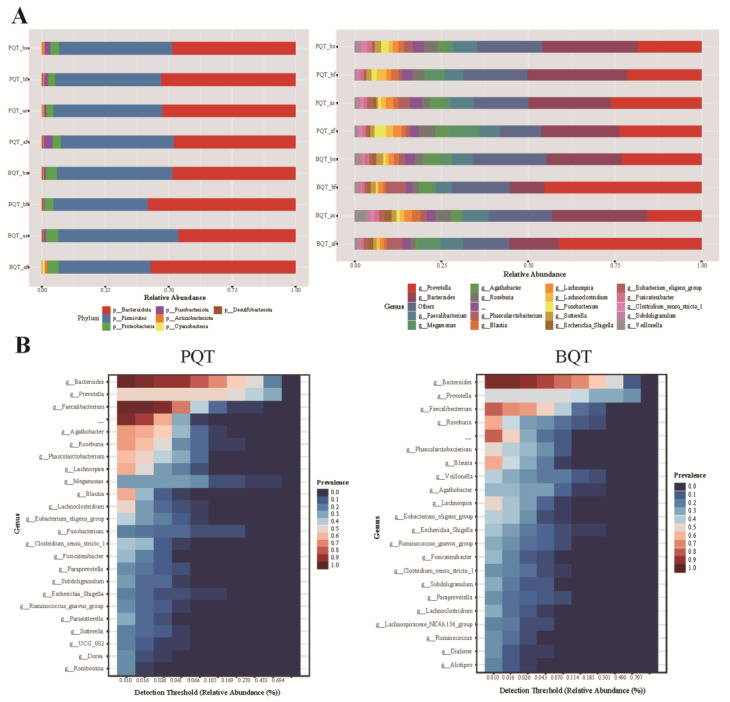
Taxa difference in gut microbiota between the PQT and BQT groups. (**A**) The relative abundance changes of phylum and genus in different groups. (**B**) The core microbiota composition between the PQT and BQT after treatment.

**Table 1 jcm-11-07050-t001:** Baseline demographic and clinical characteristics.

	Mean (SD)		*p*-Value
Characteristic	PQT (*n* = 49)	BQT (*n* = 51)	
Age (years)	51.41 (6.02)	51.24 (5.18)	0.88
Female gender, NO. (%)	27 (55)	33 (65)	0.33
BMI (kg/m^2^)	24.32 (2.35)	24.25 (2.70)	0.89
Education level, NO. (%)			0.70
Primary and below	18 (37)	21 (41)	
Junior	26 (53)	23 (45)	
Senior and above	5 (10)	7 (14)	
Place of residence, NO. (%)			1.00
City	4 (8)	4 (8)	
Countryside	45 (92)	47 (92)	
The source of potable water, NO. (%)			0.73
Well water	33 (67)	36 (71)	
Tap water	16 (33)	15 (29)	
Active smoking, NO. (%)	11 (22)	13 (26)	0.72
Alcohol consumption, NO. (%)	16 (33)	14 (28)	0.57
Tea consumption, NO. (%)	15 (31)	11 (22)	0.30
The results of gastroscopy, NO. (%)			1.00
Gastritis	49 (100)	50 (98)	
Family history of cancer, NO. (%)	10 (20)	7 (14)	0.37

PQT, polaprezinc-containing therapy; BQT, bismuth-containing therapy. BMI, body mass index.

**Table 2 jcm-11-07050-t002:** Efficacy of *H. pylori* eradication in the two groups.

Analysis	PQT (*n* = 49)	BQT (*n* = 51)	*p*-Value
ITT	30/49 (61.22%)	36/51 (70.59%)	0.32
PP	30/40 (75.00%)	36/41 (87.80%)	0.138

Data are presented as *n* (%). ITT, intention to treat; PP, per protocol; PQT, polaprezinc-containing therapy; BQT, bismuth-containing therapy.

**Table 3 jcm-11-07050-t003:** Taxa difference comparisons in gut microbiota between PQT and BQT after anti-*H. pylori* treatment.

Genus	**PQT Successful vs. BQT Successful.**	Mean Relative Abundance of PQT	Mean Relative Abundance of BQT	*p*-Value	*Q*-Value	Fold
*Parasutterella*	**>**	0.006536	0.001629	0.0007	0.045 *	4.01
*Hungatella*	**<**	0.000019	0.003719	0.0016	0.055	196.61
*Butyricicoccus*	**>**	0.005430	0.002036	0.0025	0.055	2.67
*Coprococcus*	**>**	0.006267	0.003060	0.0048	0.080	2.05
*Lachnospiraceae_UCG-004*	**>**	0.005093	0.002391	0.0076	0.102	2.13
*Christensenellaceae_R-7_group*	**>**	0.004260	0.000754	0.0102	0.114	5.65
*Subdoligranulum*	**>**	0.014834	0.005856	0.0119	0.114	2.53
*Veillonella*	**<**	0.003862	0.045077	0.020	0.168	11.67
*Lachnospiraceae_NK4A136_group*	**>**	0.008186	0.003223	0.024	0.181	2.54
*Collinsella*	**>**	0.004549	0.001708	0.038	0.20	2.66
*Anaerostipes*	**>**	0.002392	0.000751	0.040	0.20	3.19
*Agathobacter*	**>**	0.043644	0.030979	0.042	0.20	1.41

*, *q* < 0.05. PQT, polaprezinc-containing therapy; BQT, bismuth-containing therapy.

## Data Availability

The data that support the findings of this study are available from the corresponding author upon reasonable request. The datasets for this study can be found in the Harvard Dataverse [https://doi.org/10.7910/DVN/O6JZX6].

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
