# Peer review of "The Effect of Quadruple Therapy with Polaprezinc or Bismuth on Gut Microbiota after Helicobacter pylori Eradication: A Randomized Controlled Trial"

_jcm, 2022, doi:10.3390/jcm11237050_

Round 1

Reviewer 1 Report

The study's objective is to determine whether polaprezinc-containing quadruple therapy causes adverse microbiota effects among asymptomatic adults, compared with bismuth therapy. The authors conducted in-depth research on a significant issue. The topic and current status of science are stated in the introduction properly, and the findings are presented and discussed well. 

Author Response

We feel great thanks for your positive comments and suggestions. We have tried our best to polish the language in the revised manuscript. And we hope the revised manuscript could be acceptable to you.

Reviewer 2 Report

I believe that the original article entitled “The Effect of Quadruple Therapy with Polaprezinc or Bismuth on Gut Microbiota after Helicobacter pylori Eradication: A Randomized Controlled Trial” is of a high scientific quality and importance. This is even more important currently because of the dynamically spreading mechanism of antibiotic resistance of H. pylori and very limited therapeutic options.

I have some concerns covering the manuscript, including:

-          Line 72: please change the reference to the newest guidelines currently present (http://dx.doi.org/10.1136/gutjnl-2022-327745)

-          Results in the Table 2 shows that the eradication level of the tested therapy (Quadruple Therapy with Polaprezinc) is insufficient as according to the newest recommendations the level of eradication of H. pylori should be > 90% (as it is classified currently as an infectious agent) -> this information should be incorporated somehow into the manuscript (e.g. just after the sentence in lines 325-326)

-          Line 265: name of the bacterium should be written using italics

-          Please explain in detail what is the connection between the presence of Parasutterella and the development of inflammatory bowel diseases. In this particular article, the Authors point to an unfavorable relationship. So the obvious question is whether the increase in the amount of Parasutterella, that the Authors of the current manuscript put on a pedestal, after a quadruple therapy with polaprezinc is actually beneficial for health?

-     

Reviewer 3 Report

Dear authors

I read with excitement your RCT regarding H.pylori eradication with Polaprezinc vs Bismuth an its impact on gut microbiome. I found it interesting, promising for further research and generally well written. I enclose below only some minor comments, questions, in an attempt to further improve the already good manuscript. 

Genera of gut microbiota should also be written everywhere in italics, check throughout the manuscript and apply, where necessary (e.g. abstract also)

Cite the latest Guidelines of H. pylori management (Maastricht VI Consensus 2022)

The authors should clearly state whether the patients before or during the experiment received any prebiotics, probiotics or medications that might infuence gut microbiome, eg antibiotics, antidiabetics etc

Statistical analysis ïƒ  state which edition of SPSS you used, PC or Mac

After each table, please define again every used abbreviation 

Was there a difference in compliance between groups?

Regarding ethnicity were there any differences? Were they included only Chinese subjects?
